# The role of adaptation in generating monotonic rate codes in auditory cortex

**Jong Hoon Lee**[1,2]*, **Xiaoqin Wang**[1], **Daniel Bendor**[2]

**1** Laboratory of Auditory Neurophysiology, Department of Biomedical Engineering, Johns Hopkins University School of Medicine, Baltimore, Maryland, United States of America, **2** Institute of Behavioural Neuroscience, Department of Experimental Psychology, University College London, London, United Kingdom

* jlee629@jhmi.edu

**Data Availability Statement:** The source code (MATLAB) for the report's computational model is available on a Github public repository (https://github.com/dbendor/Bendor-Lab). The electrophysiological dataset is available on a

## Abstract

In primary auditory cortex, slowly repeated acoustic events are represented temporally by the stimulus-locked activity of single neurons. Single-unit studies in awake marmosets (Callithrix jacchus) have shown that a sub-population of these neurons also monotonically increase or decrease their average discharge rate during stimulus presentation for higher repetition rates. Building on a computational single-neuron model that generates stimulus-locked responses with stimulus evoked excitation followed by strong inhibition, we find that stimulus-evoked short-term depression is sufficient to produce synchronized monotonic positive and negative responses to slowly repeated stimuli. By exploring model robustness and comparing it to other models for adaptation to such stimuli, we conclude that short-term depression best explains our observations in single-unit recordings in awake marmosets. Together, our results show how a simple biophysical mechanism in single neurons can generate complementary neural codes for acoustic stimuli.

## Author summary

How is the representation of periodicity transformed in the auditory system? Previous studies have shown that stimuli comprised of repeated events are represented by precisely timed stimulus-evoked responses (temporal coding) in the early stages, and by discharge rates (rate coding) in later stages of the auditory system. In auditory cortex, a subset of neurons encodes this temporal information dually with a temporal code and a monotonically increasing or decreasing rate code. We investigated the underlying mechanisms that generate these two rate codes using a computational model of a cortical neuron. We found that pre-synaptic stimulus-evoked short-term adaptation was sufficient to generate monotonic rate codes in neurons with stimulus-synchronized activity. We validated these findings with electrophysiological data recorded from the auditory cortex of non-human primates. Together, our study suggests that a simple biophysical mechanism in single neurons can generate complex encoding and decoding of periodic stimuli.

Figshare public repository (https://doi.org/10.6084/m9.figshare.11356307.v1).

**Funding:** This research was funded by a Medical Research Council Grant (MR/M022889/1) and a European Research Council Starter Grant (CHIME) to DB, and NIH Grants DC-003180, DC-014503 to XW. The funders had no role in study design, data collection and analysis, decision to publish, or preparation of the manuscript.

**Competing interests:** The authors have declared that no competing interests exist.

# Introduction

Our ability to discriminate complex sounds such as music [1,2], speech [3,4], and conspecific vocalizations [5], relies on the auditory system's analysis of an acoustic signal's spectral and temporal structures. For sequences of brief sounds, the timing of each acoustic event is explicitly encoded by the stimulus-locked activity of neurons throughout the ascending auditory pathway. In primary auditory cortex (A1), neurons can temporally lock to individual acoustic events up to around 40–50 Hz [6–10], matching the upper limit of acoustic flutter (the percept of a sequence of discretely occurring events). While repetition rates within the perceptual range of acoustic flutter are represented by A1 neurons with stimulus-locked activity, some of these neurons can also simultaneously vary their firing rate by monotonically increasing (*Sync* +) or decreasing (*Sync-*) firing rate over the range of repetition rates that span the range of flutter perception [11]. Temporal coding provides a faithful, unambiguous representation of the timing of acoustic events. However, it must be analysed across time to determine the repetition rate of the stimulus. Rate coding, on the other hand, provides a more "processed" and instantaneous readout of repetition rate. Although rate coding is more ubiquitous in brain regions downstream from auditory cortex such as the lateral and mid-dorsal prefrontal cortex [12–16], one potential issue is that rate coding is used to represent multiple acoustic features in auditory cortex. For example, in a typical auditory cortical neuron, an increase in firing rate could represent a change in frequency, sound level [17], and/or sound location [18]. In order for rate coding to be useful to downstream brain regions, neural circuits must be able to demultiplex concurrently encoded acoustic features.

Although there are multiple ways neurons encode sensory information, in multiple brain regions including the somatosensory system, rate coding takes the form of positive and negative monotonic tuning. This form of opponent coding (positive/negative sloped rate relationship with a stimulus parameter) has been postulated to provide a number of advantages as an encoding strategy, including robustness to rate changes resulting from adaptation, allowing for the multiplexing of additional information within an overlapping rate code, and increasing the accuracy of extracting this information by reducing positively correlated noise between neurons [19]. How could the brain generate both positive and negative monotonic tuning in response to a given input? To explore this question, we used a leaky integrate-and-fire computational model of a cortical neuron. Previously, we have used a similar modelling approach to generate stimulus synchronized responses to acoustic pulses in the range of flutter perception, by varying the delay and relative strength of excitatory and inhibitory inputs [20]. In this E-I (excitation-inhibition) based computational model, synchronized responses to slowly repeating sounds occur when inhibition is both stronger than and delayed relative to excitation. Building on this model, Gao et al (2016) [21] added a simplified adaptation mechanism to stimulus repetition rate, resulting in synchronized responses and non-synchronized monotonic positive and negative responses. However, in this model stimulus repetition rate ranged beyond acoustic flutter; the integration of rate coding in synchronizing neurons, to generate Sync+ and Sync- responses within the perceptual range of flutter, has not yet been directly examined using such computational models. Here we investigated the underlying neural mechanisms responsible for Sync+ and Sync- responses in auditory cortex and demonstrate that the addition of synaptic depression to the E-I model is sufficient to reproduce these two response modes—specifically stronger synaptic depression of excitatory inputs relative to inhibitory inputs leads to the Sync- response while weaker synaptic depression of excitatory inputs relative to inhibitory inputs leads to the Sync + response.

## Results

We first examined whether the E-I model described by Bendor (2015) [20] was capable of generating both Sync+ and Sync- responses to acoustic pulse trains, using the model's three existing independent parameters: The E/I ratio (the strength of excitatory input divided by the strength of the inhibitory input), the I-E delay (the temporal lag between the excitatory and inhibitory input), and the overall strength of excitation (Fig 1A and 1B). In this model, the number of spikes produced by each acoustic event was determined by the net excitatory input. If the number of spikes produced by each acoustic event did not change with repetition rate, neurons linearly increased their discharge rate with increasing repetition rate (Sync+, Spearman correlation coefficient ρ > 0.8, P < 0.05, see methods). However, because the strength of lagging inhibition can decrease the overall net excitation in a repetition rate dependent manner, Sync- responses could be created at very high I/E ratios. While we observed that Sync+ responses were generated over a wide range of biologically plausible excitation and inhibition strengths (Fig 1C–1E), Sync- responses could only be generated using biologically unrealistic

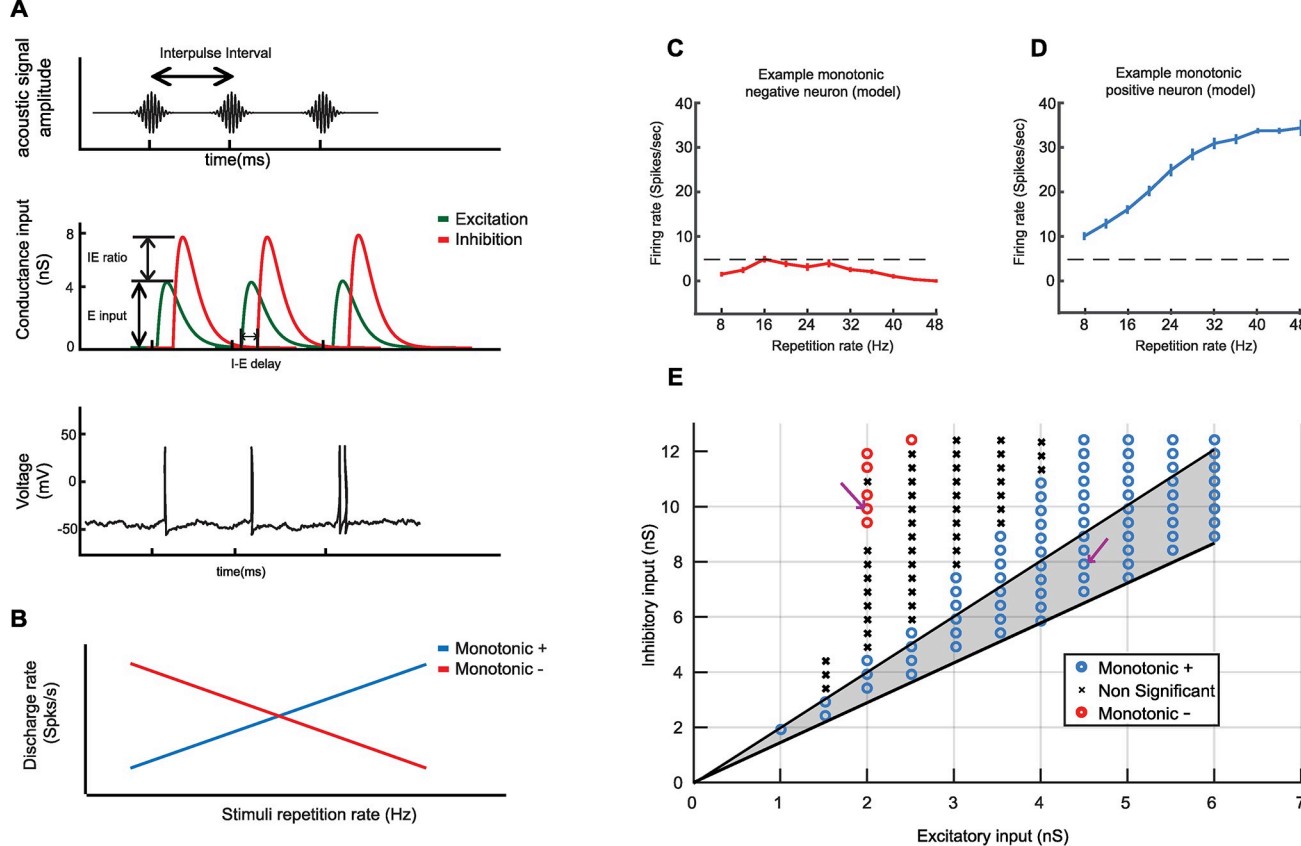

**Fig 1. Computational model of an auditory cortical neuron.** (A.) Simulated neural responses to an acoustic click train (top). each click was converted to an excitatory and inhibitory conductance input in our computational model, using an alpha function with a time constant of 5 ms (middle). Three parameters could be altered (I-E delay, E input and I/E ratio). Spikes were generated when membrane voltage reached a threshold (bottom). (B.) Cartoon of monotonic positive and negative responses. Monotonic positive neural responses increase the average discharge rate for stimuli with higher repetition rate. Monotonic negative responses decrease average discharge rate for stimuli with higher repetition rate. (C-E.) Examples of simulated neurons. Average discharge rate for increasing stimuli repetition rate for two example neurons. Model parameters for both neurons are the following: Neuron example 1 (C.): Excitatory input = 2 nS, Inhibitory input = 10 nS. Neuron example 2 (D.): Excitatory input = 4.5 nS, Inhibitory input = 8.5 nS. Error bars indicate s.e.m. (E) classification of neuron type across two parameters (Excitatory input and Inhibitory input) with a fixed I-E delay of 5 ms. The arrows indicate the parameters used for the example neurons (left arrow for example 1, right arrow for example 2). Shaded area indicates biologically plausible values where the I/E ratio is between 1.4 and 2.0.

I/E ratios; by using a 3-fold increase in the strength of inhibition relative to excitation reported in intracellular recordings [22]. Although discharge rate decreased with increasing repetition rate for these modelled Sync- neurons, their rate responses were non-significant (firing rate below 2 std above mean spontaneous rate, see methods for details.), in contrast to the driven responses observed real Sync- neurons (Bendor and Wang 2007 [11], Fig 1C).

## Modelling short-term depression

We next examined how the E-I model could be modified to more accurately represent the repetition rate tuned responses of real Sync+ and Sync- neurons. One possible mechanism that can vary discharge rate in a repetition rate sensitive manner is synaptic short-term plasticity, in particular, **short-term depression** (**STD**). If such adaptation is present, real neurons should decrease their firing rate between the start and the end of stimulus presentation. This difference would be larger for higher repetition rates, and a strong but short-term adaptation would be able to suppress the activity for high repetition rates without affecting responses for low repetition rates. Indeed, we observed that the number of spikes in real neurons at each acoustic event showed a decrease between the start and the end of stimuli sets for both Sync- and Sync + real neuron populations (Fig 2A–2C). Higher repetition rates showed a larger decrease for Sync- neurons than for lower repetition rates. The largest decrease was seen at 48Hz, the upper limit of acoustic flutter (Wilcoxon rank sum test, P ≪0.001), whereas no decrease was observed at 8Hz, the lower limit of acoustic flutter (Wilcoxon rank sum test, P = 0.1) (Fig 2D).

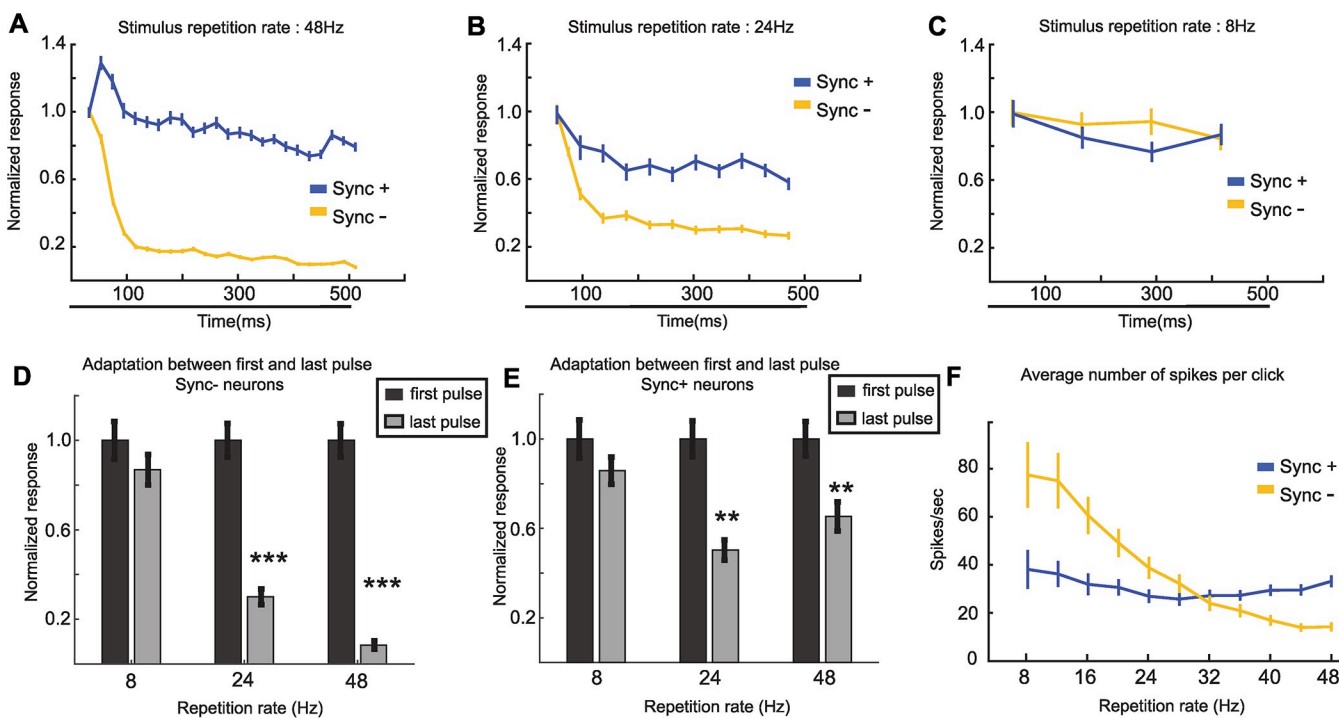

**Fig 2. Event-related activity of monotonic Sync neurons.** (A-C.) Normalized number of spikes at each acoustic event for real Sync- (n = 26) and Sync+ (n = 25) neurons at 48Hz (a.) 24Hz (B.) and 8Hz (C.). Each data point was calculated by averaging the number of spikes at the time of each acoustic event (with response latency considered). Error bars indicate s.e.m. Black bar indicates stimulus presentation period. (D-E.) adaptation between first and last acoustic event of stimulus for Sync- (D.) and Sync+ (E.) neurons. (D.) adaptation at 8Hz (Wilcoxon rank sum test, P = 0.1), 24Hz (Wilcoxon rank sum test, P ≪ 0.001), 48Hz (Wilcoxon rank sum test, P ≪ 0.001). (E.) adaptation at 8Hz (Wilcoxon rank sum test, P = 0.71), 24Hz (Wilcoxon rank sum test, P = 0.01), 48Hz (Wilcoxon rank sum test, P = 0.03). (F.) Average number of spikes of real Sync+ and Sync- neurons at each acoustic event across different repetition rates. **Sync+**: Spearman correlation coefficient = 0.36, P = 0.36; **Sync-**: Spearman correlation coefficient: 0.99, P <0.001.

Similar to Sync- neurons, the decrease was present for Sync+ neurons at 48Hz (Wilcoxon rank sum test, P = 0.03) and absent at 8Hz (Wilcoxon rank sum test, P = 0.71) (Fig 2E). When comparing this decrease between Sync- and Sync+ neurons for the same stimulus, we observed no significant difference for stimuli from 8 to 16Hz, and a significant difference from 20 to 48Hz (S1 Fig). Moreover, this depression in the neural response was stronger in the early portion of the acoustic stimulus (compared to the latter portion), and for Sync- neurons (compared to Sync+ neurons) (Fig 2A, S1 Fig). Sync+ neurons showed a weak global depression throughout stimulus presentation, and the profile of depression was not affected by repetition rate (Fig 2A, S1 Fig). Finally, the average number of spikes per acoustic event decreased monotonically (Spearman correlation coefficient: 0.99, P <0.001) for higher repetition rate in Sync- neurons, but not in Sync+ neurons (Spearman correlation coefficient = 0.36, P = 0.36.) (Fig 2F). Together, these observations suggest that adaptation to repeated stimuli was stronger for Sync- neurons than for Sync+ neurons (S2 and S3 Figs).

## Model parameters

To add short-term depression to our previous model, we introduced two additional parameters; the amplitude of depression ($A_D$) which determined the strength of adaptation after each acoustic pulse, and the time constant of recovery ($\tau_P$) which controlled how stimulus repetition rate affected adaptation during stimulus presentation[23,24] (Fig 3A)(see methods for details). To control the strength of depression in our modified E-I model, we independently varied these two parameters for both excitatory and inhibitory inputs. We observed that by varying these two parameters, we were able to produce Sync+ (Spearman correlation coefficient $\rho > 0.8$, P < 0.05) and negative ($\rho <$ -0.8, P < 0.05) responses (Fig 3B–3D, see Methods). To further study the effects of these parameters, we first calculated the probability of obtaining monotonic positive (Fig 3B) or negative (Fig 3C) neurons across all values of $A_D$ for a given set of time constants {$\tau_{pE}, \tau_{pI}$} within a naturalistic range (between 0.05s and 0.2s) [24, 25]. This was determined so that with values in the middle of the range, neurons would show no or very little depression for repetition rates under 8Hz, which corresponded to a time interval greater than 0.125s between two pulses. The average monotonicity index of model neuron responses across all values of $A_D$ was highest for high $\tau_{pI}$ and low $\tau_{pE}$ values, and lowest for low $\tau_{pI}$ and high $\tau_{pE}$ values (Fig 3B and 3C). For a given set of time values { $\tau_{pE}$ = 0.15s, $\tau_{pI}$ = 0.10s} we were able to obtain Sync+ neurons with strong depression of inhibitory inputs and weak adaptation of excitatory inputs. The converse was true for Sync- neurons, where the strength of depression was stronger for excitation than inhibition (Fig 3D). In our parameter range, depression of excitation was more important than depression of inhibition in determining whether a neuron would be monotonic positive or negative. In this computational model, as in the previous model [20], the initial onset response was determined by the strength of excitation and inhibition, but not affected by synaptic depression. Values for excitatory and inhibitory input were chosen so that the onset response was on average between 40 and 60 spikes per second to match onset responses observed in real neurons [11], although different amplitudes of onset response did not affect our observations (S4 Fig).

For our simulated neurons, $A_D$ values were determined (Fig 4A) so that simulated Sync + and Sync- neurons matched real neurons in both trial-by-trial spiking activity (Fig 4) and average population activity (S5 Fig). Out of 107 real synchronized neurons, 25 neurons were classified as Sync+ (Fig 4C) and 26 were classified as Sync- (Fig 4E). Both simulated (Fig 4B and 4D) and real (Fig 4C and 4E) neurons showed stimulus-synchronised driven activity to stimuli, with both simulated (Fig 4D) and real Sync- neurons showing a reliable adaptation of firing rate to higher stimulus repetition rates. Monotonicity was significant for both Sync+

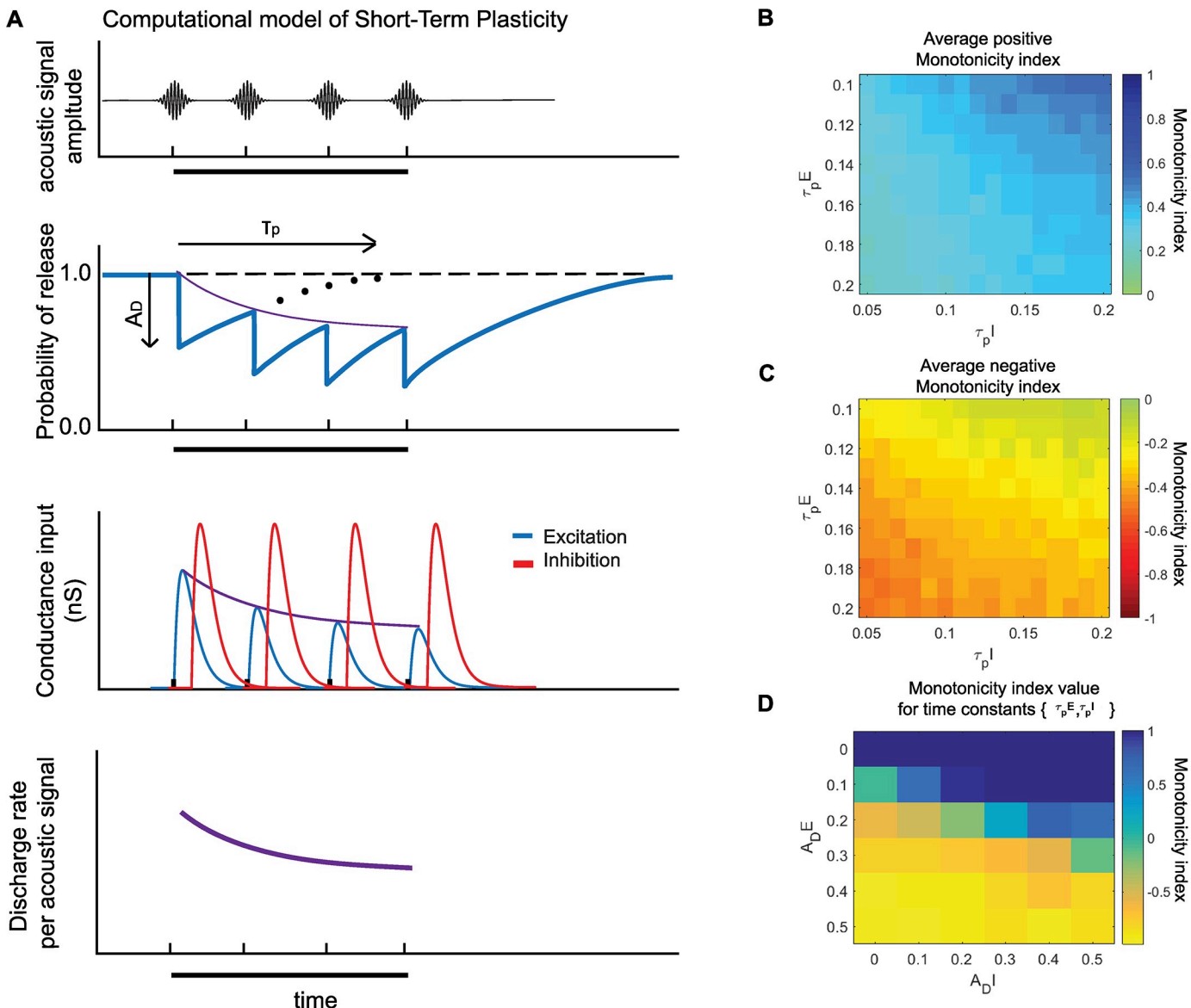

**Fig 3. Computational model of an auditory cortical neuron with short term depression.** (A.) At each acoustic signal (top) we simulate the decrease in the probability of release of synaptic vesicles with an amplitude of adaptation $A_D$ followed by an exponential recovery with time constant $\tau_p$ (middle top) (See methods for details). This probability of release then determined the amplitude of conductance input to our model neuron (middle bottom). a decrease in conductance amplitude during stimuli presentation (black bar) resulted in a decrease in discharge rate per acoustic signal (bottom). (B-D.) Adaptation parameter space. Average positive (B.) and negative (C.) monotonicity index for a given set of recovery time constants $\{\tau_{pE}, \tau_{pI}\}$. Average monotonicity index at $\{\tau_{pE} = 0.15, \tau_{pI} = 0.10\}$ for different values of $A_{DE}$ and $A_{DI}$ (D.).

(Spearman correlation coefficient $\rho = 0.91$, $P < 0.001$) and Sync- ($\rho = 0.85$, $P = 0.012$) simulated neurons (Fig 4F and 4G), and temporal fidelity over the range of repetition rates spanning flutter perception was maintained despite adaptation (Fig 4B–4E, Vector strength (VS) > 0.1, and Rayleigh statistic > 13.8, $P < 0.001$).

## Model robustness

Although our simulated neurons show similar Sync+ and Sync- responses to averaged real Sync+ and Sync- responses, we observed that in real neurons, innate properties such as

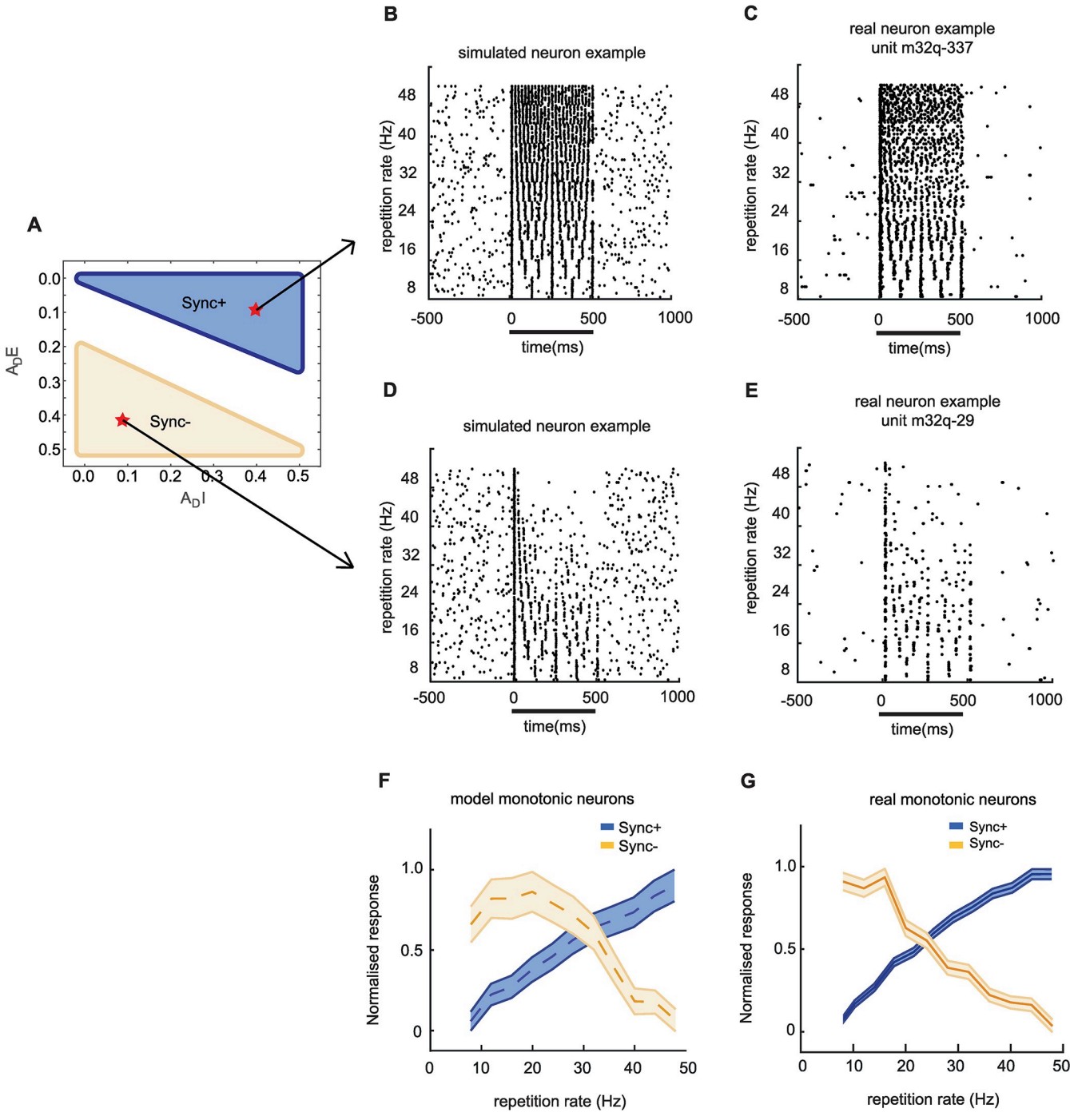

**Fig 4. Real and Simulated monotonic Sync example neurons.** (A.) Range amplitude of adaptation $\{A_{DE}, A_{DI}\}$ that results in Sync+ and Sync- simulated responses. Raster plot comparison between simulated Sync+ (B; $A_{DE} = 0.4$, $A_{DI} = 0.1$, $\tau_{pE} = 0.15$ s $\tau_{pI} = 0.10$ s.) and Sync- (D; $A_{DE} = 0.1$, $A_{DI} = 0.4$, $\tau_{pE} = 0.15$ s $\tau_{pI} = 0.10$ s.) neurons with real Sync+ (C; unit m32q-337) and Sync- (E; unit m32q-29) neuron examples. the black bar indicates the time during when stimuli was given as input. $A_{DE} = 0.4$, $A_{DI} = 0.1$, $\tau_{pE} = 0.15$ s $\tau_{pI} = 0.10$ s. (F,G.) Normalized discharge rate for Sync + and Sync- neurons across stimuli with different repetition rates. Discharge rate was normalized to the maximum value across stimuli. (F.) Population average of simulated Sync+ and Sync- neurons. (G.) Population average of real Sync+ and Sync- neurons.

spontaneous rate and onset response to stimuli varied among neurons within the same category. We therefore examined whether the model's ability to produce Sync+ and Sync-

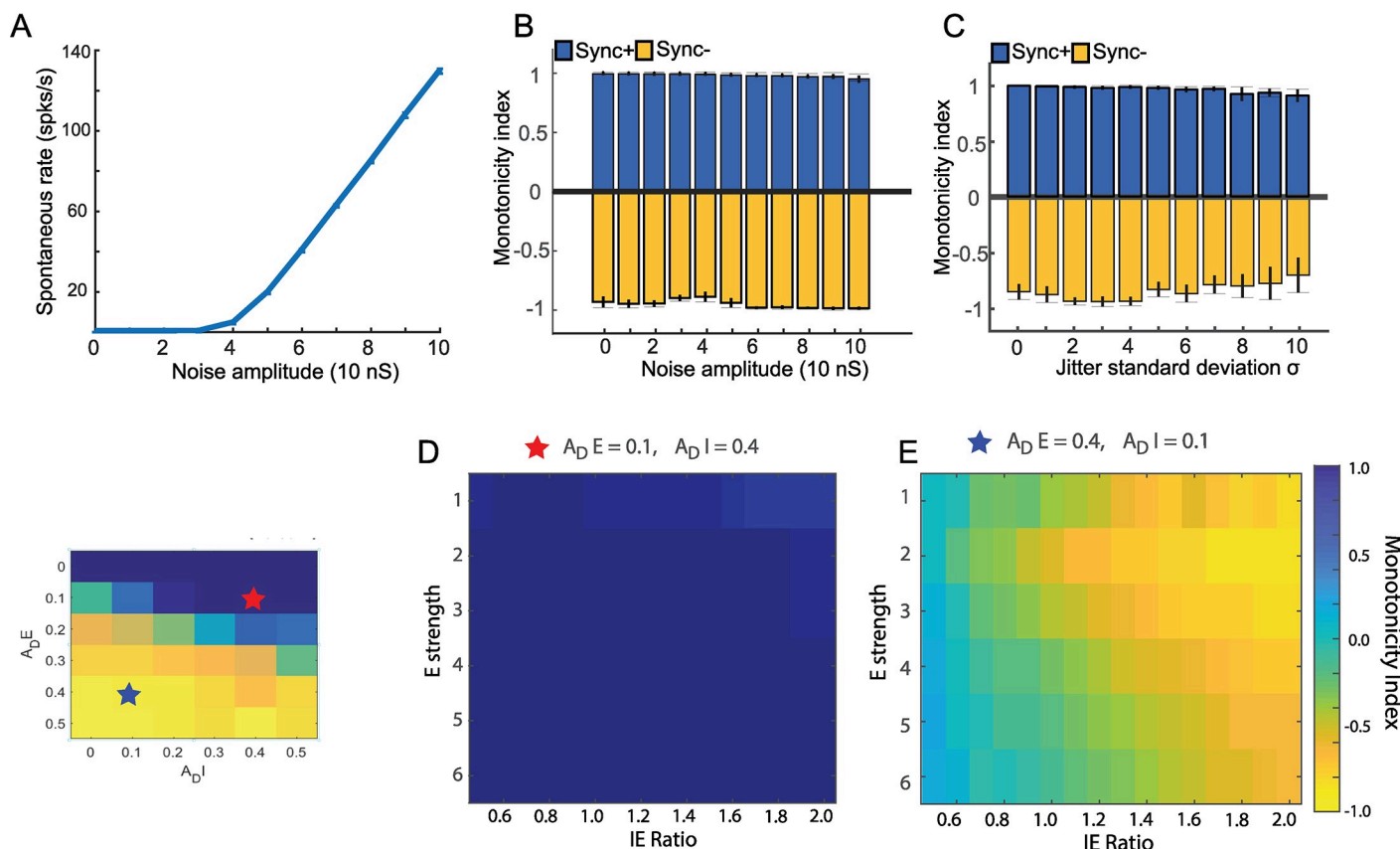

**Fig 5. Model robustness.** Spontaneous rate in relation to noise amplitude (A.) Monotonicity in Sync+ and Sync- neurons in relation to noise amplitude (B.) and temporal jitter (C.) Average monotonicity index (D, E.) across different values for recovery time constants $\{\tau_{pE}, \tau_{pI}\}$ ranging between 0.06 and 0.20s for a given value of $\{A_{DE}, A_{DI}\}$. When within the parameters of producing Sync+ neurons, Monotonicity is unaffected by changes in E strength and IE ratio (A.). For parameters resulting in Sync- neurons, Monotonicity is negative only when inhibition is stronger than excitation (IE ratio larger than 1) (B.).

simulated neurons were robust to changes in initial conditions of the model. For this purpose, we first examined the robustness of our model to different types of noise. Our computational model operated, as did the previous model [20], with a fixed spontaneous rate (~4 spk/s) comparable to that of our real neuron data (median spontaneous rate = 3.8 spk/s). This was generated by adding Gaussian noise in addition to the excitatory and inhibitory conductances of the neuron (see methods). Increasing the amplitude of noise also increased the spontaneous rate (Fig 5A). We examined how robust our model was for varying noise amplitude and observed that it did not affect monotonicity for both Sync+ and Sync- simulated neurons (Fig 5B). Vector strength was less robust to changes in noise amplitude, in particular for Sync- simulated neurons, where low noise amplitude resulted in a complete lack of stimulus synchrony for high repetition rates (S6 Fig), due to the evoked responses consisting of an onset followed by suppression. Our model also included temporal jitter (Fig 5C) to emulate more realistic responses, by adding Gaussian noise to the timings of each acoustic pulse. Similar to the conductance noise amplitude, changes to temporal jitter did not affect monotonicity. We also observed that the vector strength in Sync- simulated neurons was more affected by temporal jitter than for Sync+ simulated neurons (S6 Fig). However, with the exception of Sync- simulated neurons with strong temporal jitter (above 7 s.d.) these simulations in the presence of noise could still be classified as synchronised monotonic responses (see Methods for criteria). We further explored model robustness by studying how input parameters such as excitation

and inhibition amplitude affected monotonicity and vector strength. Monotonicity in Sync + simulated neurons did not seem to be affected by changes in these parameters (Fig 5D). In Sync- neurons however, the monotonicity index was reduced to 0 for IE ratios under 1.0 (Fig 5E). In addition, for stronger excitatory input amplitudes the model required higher IE ratios to produce monotonic negative responses. As for vector strength, both Sync+ and Sync- simulated neurons showed a weak decrease in stimulus synchrony for excitatory input amplitudes under 2nS (S6 Fig).

### Different mechanisms for adaptation to repeated acoustic pulses

So far in this study we explored short-term depression as a possible underlying mechanism for Sync+ and Sync- neurons observed in A1. Next, we explored other possible mechanisms that may allow neurons to adapt to acoustic pulse trains and compared their effects to that of our short-term depression model. One such mechanism is **short-term facilitation (STF)**; the adaptation of neural activity during stimulus presentation for higher repetition rates could arise from facilitation of inhibition, as opposed to depression of excitation. We thus modelled short-term facilitation using the same parameters as short-term depression. However, instead of decreasing the probability of release (and therefore the conductance input amplitude), this probability was increased at each acoustic input until it was recovered back to its initial value (Fig 6) (see methods). When combining depression of excitation and facilitation of inhibition, the model was able to produce both Sync+ and Sync- responses. Similar to our original model (depression of excitation and inhibition) depression of excitation was the determining factor for the direction of monotonicity for simulated neurons (Fig 6A and 6B). However, increasing the strength of facilitation in the inhibitory input lead to a decrease in the monotonicity slope of Sync+ neurons and an increase in the monotonicity slope of Sync- neurons. When both excitation and inhibition were facilitated, all simulated neurons were Sync+ neurons (Fig 6C and 6D). In the case where there was strong facilitation of inhibition and weak depression of excitation, our model produced non-monotonic synchronized responses (highest discharge rates in the middle of the acoustic flutter range).

Another possible mechanism for adaptation to stimulus statistics is **spike-frequency adaptation (SFA)**. Although the time scale for SFA is generally much shorter than that of short-term depression [26, 27], the two effects could be complimentary. In order to separate SFA from our observations, we studied Inter-Spike Intervals (**ISIs**) at onset for both Sync+ and Sync- real neurons by comparing the difference between the first and second ISI and second and third ISI (S7 Fig). Within the same population, we observed a significant difference between the first, second and third ISI (KS test, P <0.05), and thus the presence of SFA. However, the time scale of SFA was in the order of 0.5ms, compared to the time scale of flutter (20 to 125 ms). In addition, SFA at the onset between Sync+ and Sync- neurons was significantly different (t-test, P<0.05) but the difference was in the order of 1ms.

To further compare the aforementioned mechanisms between each other and with real neuron populations, we studied the strength of adaptation in relation to discharge rate at different time windows during the stimulus presentation (acoustic pulse train with a repetition rate of 40Hz). The strength of adaptation, equivalent to the amplitude of adaptation $A_D$ shown in the model above, was defined as the firing rate during the time window spanning the given acoustic pulse divided by the firing rate during the previous acoustic pulse. Real neurons with firing rates lower than the spontaneous rate during the first 2 pulses (5/26 neurons in Sync-, 7/25 neurons in Sync+) were excluded from analysis. The strength of adaptation was also calculated for models with different mechanisms for adaptation; the base model with STD for excitation and inhibition, the base model with additional weak or strong SFA (see methods), and

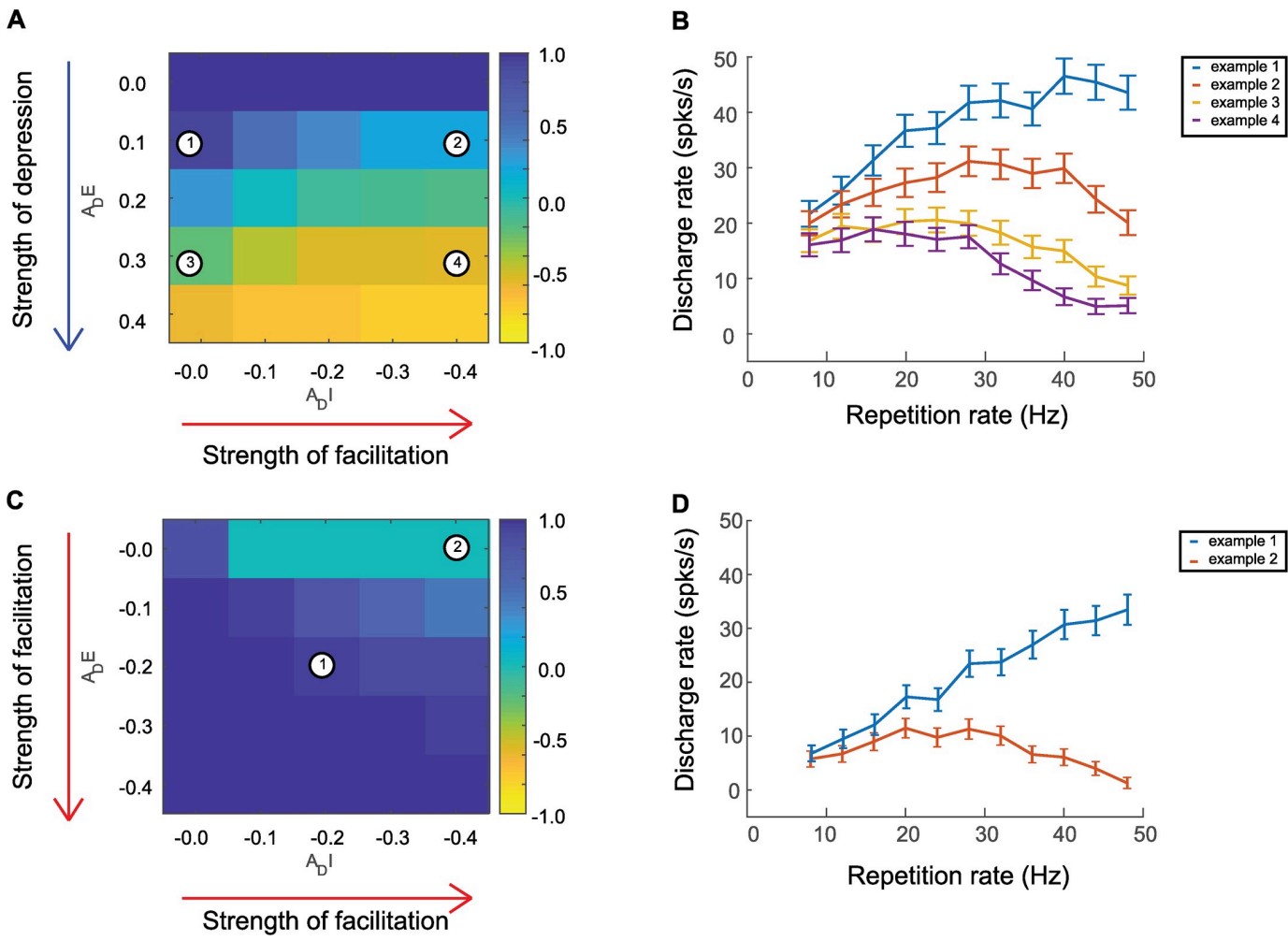

**Fig 6. Computational model including short term facilitation.** Short term facilitation was added to the model by increasing the probability of release (see methods) at each acoustic input, which would decay back to the initial value with a time constant $\tau_p$. Initial probability of release was 0.5 compared to 1.0 in short term depression model to compensate for changes in conductance input amplitudes. (A, B.) Depression of excitation and facilitation of inhibition. (C, D.) Facilitation of both excitation and inhibition. (A, C.) average monotonicity index for a given value of adaptation amplitude $A_D$, for time constants $\{\tau_{pE}, \tau_{pI}\}$ ranging between 0.06 and 0.20s. (B, D.) discharge rates for example neurons. (B.) Simulated neurons with depression of excitation and facilitation of inhibition. Example neuron 1 at $\{A_{DE} = 0.1, A_{DI} = -0.0\}$, Spearman correlation coefficient = 0.76, P = 0.01. Example neuron 2 at $\{A_{DE} = 0.1, A_{DI} = -0.4\}$, Spearman correlation coefficient = 0.25, P = 0.45. Example neuron 3 at $\{A_{DE} = 0.3, A_{DI} = -0.0\}$, Spearman correlation coefficient = -0.66, P = 0.03 Example neuron 4 at $\{A_{DE} = 0.3, A_{DI} = -0.4\}$, Spearman correlation coefficient = -0.74, P = 0.01 (D.) Simulated neurons with facilitation of both excitation and inhibition. Example neuron 1 at $\{A_{DE} = -0.2, A_{DI} = -0.2\}$, Spearman correlation coefficient = -1, P << 0.001. Example neuron 2 at $\{A_{DE} = -0.0, A_{DI} = -0.4\}$, Spearman correlation coefficient = 0.07, P = 0.84. Time constants of all example neurons: $\{\tau_{pE} = 0.15, \tau_{pI} = 0.10\}$.

the facilitation model with STD for excitation and STF for inhibition (Fig 7A). As expected, adaptation during the first to second pulse for Sync- simulated neurons was strongest in the strong SFA model, and weakest in the facilitation model. Adaptation increased significantly between first to second pulse and first to third pulse for the base model and for the facilitation model (Wilcoxon sign rank test P <<0.001) but not for models with weak or strong SFA (Wilcoxon signed rank test P = 0.06 and P = 0.5 respectively). For Sync+ neurons, all models showed a weak or non-significant adaptation. In the case of real Sync- neurons, most neurons showed significant depression between the first and second pulse (18/21 neurons, median = 0.59, t-test, P << 0.001) and between first and third pulse (18/21 neurons, median = 0.90, P << 0.001) (S8 Fig), and the difference of adaptation strength between these two time

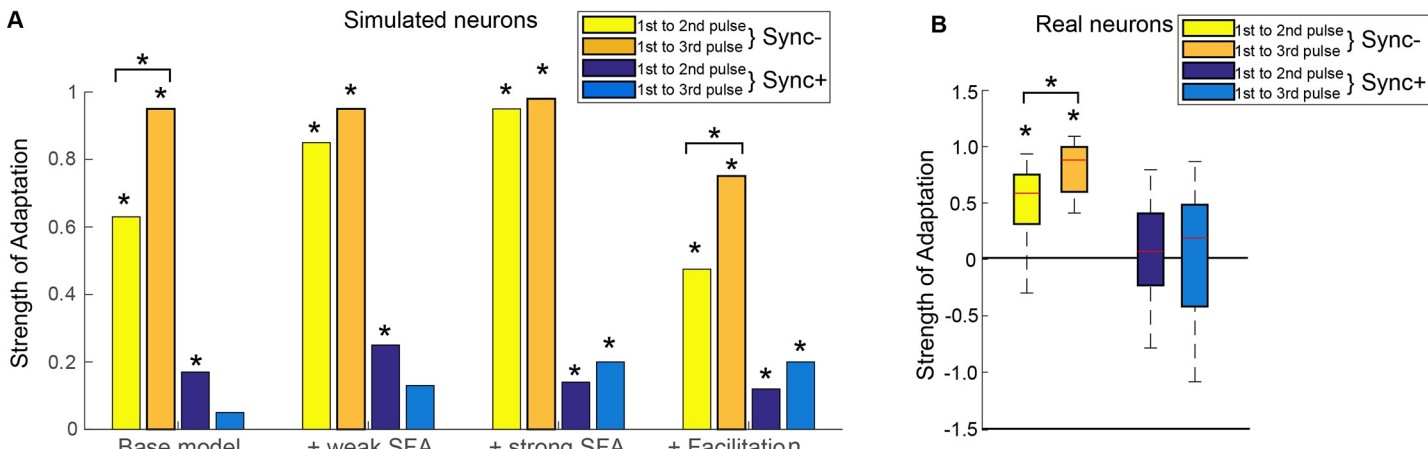

**Fig 7. Adaptation between individual acoustic events for real neurons and different models.** (A.) Adaptation between the 1st and 2nd input, and between 1st and 3rd input for Sync+ and Sync- neurons for models with different adaptation mechanisms. Strength of adaptation increased significantly between 1st to 2nd pulse and 1st to 3rd pulse for Sync- base and facilitation model (Wilcoxon signed rank test P << 0.001) (B.) Strength of adaptation in real Sync+ and Sync- neurons between the 1st and 2nd input, and between 1st and 3rd input. Strength of adaptation increased significantly between 1st to 2nd pulse and 1st to 3rd pulse for Sync- neurons (Wilcoxon signed rank test P <0.01). Asterisks directly above bars indicate that the adaptation amplitude was significantly different from 0 (Wilcoxon signed rank test P <0.05).

windows was statistically significant (Wilcoxon rank sum test, P < 0.01) (Fig 7B) these results were most comparable to our base model using only short-term depression. As for Sync+ neurons, individual responses showed both depression and facilitation during onset. 9/18 neurons and 8/18 neurons showed depression between 1st and 2nd pulses and between 1st and 3rd pulses respectively (median = 0 for both, Wilcoxon signed rank test, P > 0.05) (Fig 7B). Using the same methods, we also calculated the strength of adaptation for synchronized non-monotonic (SyncNM) neurons (see methods). Adaptation strength for SyncNM neurons were weaker than that of Sync- neurons (53/55 neurons, median = 0.25, P <<0.001), and closely matched our base models' parameters needed to produce SyncNM simulated neurons. Among the three sub populations, adaptation strength was significantly different between Sync- neurons and SyncNM neurons (Wilcoxon rank sum test, P <0.01) and between Sync- neurons and Sync+ neurons (Wilcoxon rank sum test, P <0.01), but not between SyncNM and Sync + neurons (Wilcoxon rank sum test, P = 0.44). These results showed that short-term depression was sufficient to reproduce adaptation to acoustic pulse trains in all auditory cortical neurons (Sync+, Sync- and SyncNM) synchronized to acoustic flutter.

## Response to pure tones

If we consider pure tones to be similar to acoustic pulse trains with a very high repetition rate, the responses these stimuli evoke in Sync+ neurons and Sync- neurons would be different. We would more likely observe a brief onset response in Sync- neurons compared to a more sustained response observed in sync+ neurons. Using our computational model, we could also emulate responses of Sync+ and Sync- neurons to different sets of stimuli such as pure tones. In real neurons, similar responses were evoked by pure tones (at the neuron's best frequency) and pulse trains with high repetition rates (Fig 8A). We observed an onset followed by a damped sustained response for Sync+ neurons and an onset followed by a suppressed response for Sync- neurons. Both our computational model for Sync+ and Sync- neurons behaved similarly to real neurons (Fig 8B), with Sync- simulated neurons showing a transient onset followed by a suppressed response, whereas Sync+ showed a damped sustained response during stimulus. Our simulated responses to pure tones did however differ from real neuron response

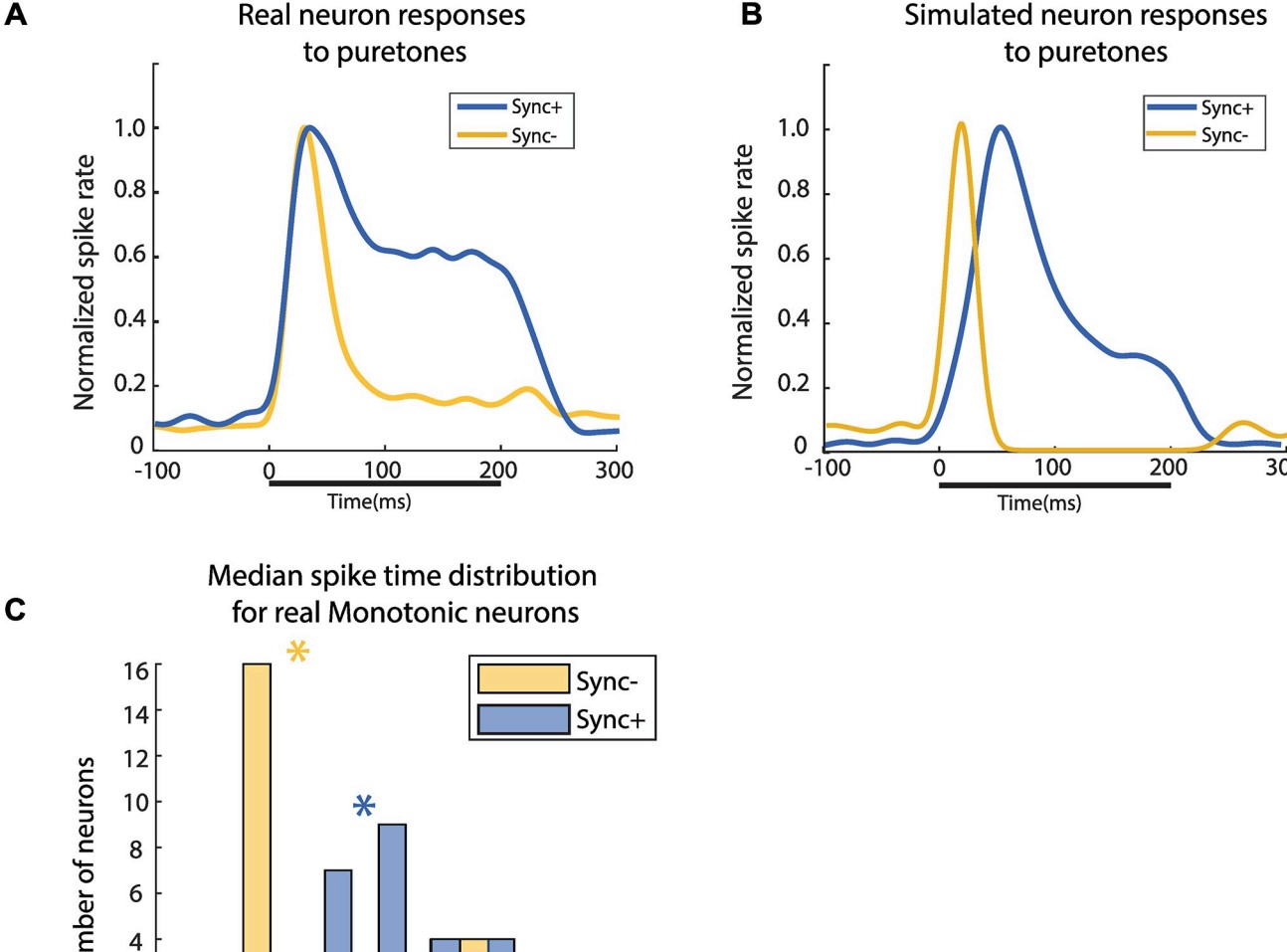

**Fig 8. Pure tone responses.** Normalized responses to pure tones in real (A; Sync +, n = 25. Sync-, n = 26) and simulated neurons (B; 30 simulated neurons). Normalized spike rate was obtained by dividing the population average response to the average peak response during stimulus presentation. (C.) Distribution of median spike times during stimuli presentation of all Sync+ neurons (green asterisk: median of distribution = 89ms) and Sync- neurons (yellow asterisk: median of distribution = 44ms). The two distributions were significantly different (Wilcoxon rank sum test, P < 0.001).

dynamics (S9 Fig). Sync+ responses were greatly exaggerated in our simulated neurons compared to real neurons, with the peak response time being significantly later than onset response time. Decreasing the initial excitatory input amplitude or introducing SFA to the model seem to affect Sync- responses, however increasing the excitation strength led to a proportional increase in onset response (S10 Fig). These data suggest that the temporal profile of pure-tone responses could be used to predict whether a neuron is Sync+ or Sync-, even though actual firing rates of the base model did not accurately reflect real neuronal responses. We tested this prediction by measuring the median of all spike times during stimulus presentation of pure tone responses in real and simulated neurons: Neurons with sustained responses would have a higher median spike time during stimulus presentation than those showing onset responses. This was indeed the case for both real neuron populations (Fig 8C) (median spike time of Sync + neurons = 89ms, median spike time of Sync- neurons 44ms, Wilcoxon rank sum test,

P < 0.05) and simulated neurons (S10 Fig). Results for Sync+ simulated neurons suggest that median spike times during the stimulus varies depending on the strength of adaptation whereas for Sync- neurons it stays constant. This could explain the greater variation of median spike times in real Sync+ neurons compared to Sync- neurons (Fig 8C).

## Discussion

Here we describe a computational model able to reproduce the monotonically-tuned synchronized responses of auditory cortex neurons evoked by acoustic pulse trains in the range of flutter perception. By adding the parameters of pre-synaptic short-term depression to both the excitatory and inhibitory inputs of the initial conductance-based integrate-and-fire E-I model [20] we were able to model both positive and negative monotonically tuned, stimulus synchronized neurons (Sync+ and Sync-). Sync+ responses were generated when adaptation for excitation was weak or not present, whereas Sync- responses were generated when adaptation for excitation was strong. Adaptation of inhibition played a role in facilitating or supressing post-synaptic responses. This adaptation was modelled using a realistic set of time constants and rates of adaptation, consistent with previous studies across multiple laboratories [6,28–32]. When compared with other possible mechanisms for adaptation such as pre-synaptic short-term facilitation and post-synaptic spike-frequency adaptation, our model best emulated adaptation of real Sync+ and Sync- neurons to acoustic pulse trains in the perceptual range of flutter and was also able to make testable predictions such as temporal dynamics of responses to pure-tones, which was subsequently confirmed in our real neuronal population. In addition, our findings suggest that synaptic depression of excitation may play a role in diversifying how the auditory cortex encodes stimuli. This observation was in line with a recent study [33] which demonstrated an important role of synaptic depression of excitation during natural sound processing in the ferret auditory cortex.

Another possible mechanism for adaptation to repeated stimuli is stimulus specific adaptation (SSA) which has been widely studied in the auditory cortex in mammals [34–38]. Although we did not model the effects of SSA in this study, previous groups have shown that the timescale for SSA to repetitive stimuli is much slower than the rate of flutter (0.25Hz to 8Hz) [35,38,39,40]. This led us to believe that Sync+ and Sync- responses found in auditory cortex were not a result of SSA.

If spike timing already contains stimulus relevant information, what could be the functional role of Sync+ and Sync- responses? Although primary somatosensory [8] and auditory cortices encode stimulus timing using both a rate and temporal representation, downstream neurons may only be processing one of these inputs. Mountcastle and colleagues [41] previously postulated that a neural mechanism could read out the periodic inter-spike intervals of the spike trains evoked in S1. In anesthetized animals, ISI does contain by far the highest amount of information, assisted by information from firing rate [42]. However multiple studies in awake animals [43–46] in both sensory areas have shown that firing rate, not precise spike timing, more accurately represents the psychophysical discrimination thresholds of stimulus repetition rate.

In addition, we observe a loss of temporal fidelity to repetitive stimuli as we move along in the auditory pathway from the auditory periphery to cortex (e.g., cochlear nucleus: [47–50], inferior colliculus: [51–53] medial geniculate body: [29,54], auditory cortex: [8,9,55–57]) due to biophysical properties of neurons and temporal integration of converging inputs from one level to the next [49]. This loss of temporal fidelity in the auditory cortex, while problematic for a temporal representation, is mitigated by the substitution of a rate code for encoding the same information. Rate-based representation to repetition rate represent processed temporal

information, as opposed to preserved temporal information in ISI based representations. With rate coding, cortical processing of auditory streams would operate on a "segment-by-segment" basis, and not on a "moment-by-moment basis" as found in the auditory periphery [9,58]. As previous studies [11,21] have suggested, this would allow temporal integration of information over specific time intervals, which are required by higher-level processing tasks. The transformation from temporal to rate code would be necessary for such complex cortical integration.

Thalamic and prethalamic areas in the auditory pathway contain predominately synchronized neurons, while non-synchronized (nSync) neurons using firing rate to encode temporal information for repetition rates above the upper limit of flutter are most prevalent in auditory cortex (and to a limited extent the medial geniculate nucleus (MGB)) [49]. Both Sync (temporal coding) and nSync neurons (rate coding) responding to flutter were found in A1 and in the Rostral fields (R and RT), although a higher proportion of Sync+/- neurons were found in A1, compared to R and RT where there were more nSync+/- neurons (monotonically encoding repetition rates within the range of flutter perception). Similar transformations were found in the Somatosensory pathway from Thalamus to S1, S2 [29] where, in the same manner as the auditory cortex, S2 neurons showed a much weaker stimulus-locking than S1 for vibrotactile stimuli and encoded temporal information using either positive or negative monotonic rate codes. Based on these observations, we postulate that Sync+/- neurons are an intermediary stage in the transformation of stimulus information encoding from a temporal representation to a rate code lacking stimulus locked responses. Previous studies have suggested single-compartment computational models to explain this transformation of temporal encoding across the auditory system [20,28,59,60], but all of these studies have grouped synchronized neurons in a single rate-coding category, not distinguishing between positive/negative monotonic neurons. One study in particular [21] proposed a similar model incorporating short-term depression to produce Sync+ and Sync- responses. Despite the similarities in the approach, the model described in this study differs from the current model in two major ways. First, the previous model deviated from a widely used STD model (Tsodyks et al) by reducing the probability of release (see methods) to 0 each time a spike is occurred. Whereas the STD model by Tsodyks and colleagues has been fitted to and compared with real data [6,28–32], there are currently no previous studies supporting the more simplified model. Second, we define Sync + and Sync- neurons as neurons having monotonic positive or negative, synchronised responses within the perceptual range of flutter, as seen in real neurons [11]. The previous model calculates monotonicity across the perceptual ranges of both flutter and fusion, but the resulting Sync- and nSync- simulated neurons do not seem to be monotonically negative within the range of flutter. Therefore, this model is not accurately modelling real neurons observed in Bendor and Wang 2007 [11]. We decided to improve upon these two areas to make the model more realistic and more accurately emulate Sync+ and Sync- neurons observed in previous data. Additionally, we do not vary the temporal integration window of neurons (a key model parameter in Gao et al. 2016), as there is a limited experimental data defining the biologically realistic upper range in auditory cortex.

There are, however, several caveats to our computational model. First, we compare single unit data from marmosets with simulated neurons using cellular parameters based on intracellular recordings of ketamine-anesthetized rats [28], due to the fact that no such data exists for marmosets. Because ketamine is an NMDA antagonist, our model only simulated AMPA and GABA-A receptors, making no distinction between the two. NMDA receptors produce synaptic inputs with a longer time-constant (10-25ms) than AMPA and GABA-A receptors (5ms) and may thus explain the difference in response between awake and anesthetized animals. Previous studies have introduced NMDA receptors to single compartment models [20,59,60], but none have studied how it affects the monotonicity of synchronized responses.

Second, our model does not account for different sources of inhibition–such as top-down inhibitory feedback or cortical inhibition from different interneurons—that could affect excitatory pyramidal cells in the auditory cortex. In particular, there has been increasing evidence of top-down signals mediating rapid, behaviourally driven modulatory effects in the auditory cortex [46, 61–63]. A recent study has shown that a network of Parvalbumin-positive (PV) inhibitory neurons may mediate such top-down control by rapidly and efficiently modify sensory responses in auditory cortex in mice [63]. However, the timescale of adaptation occurring in Sync+ and Sync- neurons was within the order of milliseconds, and therefore the contribution of such top-down inhibition seemed unlikely. In addition, the spontaneous rate of cortical neurons did not decrease for stronger adaptation as it did with activation of PV inhibitory neurons in the mentioned study. Nonetheless, we cannot rule out possible effects of inhibition from other interneurons with shorter, more transient timescales as observed in somatostatin (SOMs) and vasoactive intestinal polypeptide-expressing (VIPs) interneurons [64,65]. Further studies using optogenetics to manipulate different types of inhibitory neurons may provide additional insight on how inhibition affects adaptation in the auditory cortex.

Alongside acoustic pulse trains, Bendor and Wang [11,58,66] also recorded responses of the same neurons to sinusoidal amplitude modulated (SAM) tones and to pure tones. In the current model, an acoustic pulse is modelled as a single excitatory gaussian kernel followed by an inhibitory kernel. SAM tones have different spectral bandwidth and pulse duration depending on the modulation frequency [66] and cannot be represented accurately by our model. As for pure tone responses, our model represents the input as a net onset excitation followed by inhibition during stimulus presentation. Our model also considers that A1 neurons receive the same excitatory and inhibitory conductance input for each acoustic pulse regardless of the repetition rate. However, both in vitro [67] and in silico [59] studies show evidence for short-term plasticity to repetitive acoustic stimuli for projections from Inferior Colliculus (IC) to MGB neurons. Inputs to A1 neurons originating from acoustic pulses would have therefore passed such filters. While the addition of parameters that account for these different types of stimuli and transformations could provide further improvements to the model, our aim was to demonstrate that the addition of adaptation to a simple computational model is sufficient to produce positive and negative monotonic rate coding in stimulus-synchronizing neurons.

## Materials & methods

### Ethics statement

The electrophysiology data used in this study comprised of a previous published dataset [11] collected in the laboratory of Professor Xiaoqin Wang at Johns Hopkins University. All experimental procedures were approved by the Johns Hopkins University Animal Use and Care Committee and followed US National Institutes of Health guidelines.

### Electrophysiological recordings and acoustic stimuli

Our electrophysiology data in this report comprised previous published datasets [11]. For these datasets, the authors performed single-unit recordings with high-impedance tungsten micro-electrodes ($2–5M\Omega$) in the auditory cortex of four awake, semi-restrained common marmosets (Callithrix jacchus).

Action potentials were sorted on-line using a template-matching method (MSD, Alpha Omega Engineering). Experiments were conducted in a double-walled, soundproof chamber (Industrial Acoustic Co., Inc.) with 3-inch acoustic absorption foams covering each inner wall (Sonex, Illbruck, Inc.).

Acoustic stimuli were generated digitally (MATLAB- custom software, Tucker Davis Technologies) and delivered by a free-field speaker located 1 meter in front of the animal. Recordings were made primarily for the three core fields of auditory cortex- primary auditory cortex (AI), the rostral field (R), and the rostrotemporal field (RT), with a subset of neurons recorded from surrounding belt fields. For each single unit isolated, the best frequency (BF) and sound level threshold was first measured, using pure tone stimuli that were 200 ms in duration. We next generated a set of acoustic pulse trains, where each pulse was generated by windowing a brief tone at the BF by a Gaussian envelope. Pulse widths ranged from $\sigma = 0.89$ to 4.65ms. Repetition rates ranged from 4Hz to 48Hz (in 4Hz steps). The pulse train stimuli were 500ms in length, with at least a 500ms pre-stimulus period and a 500ms post-stimulus period. The number of repetitions for each stimulus was at least five, and at least ten for most of the neurons (236/274). Stimuli was presented in a random shuffled order, and intensity levels were generally 10 – 30dB above BF pure tone thresholds for neurons with monotonic rate-level functions and at the preferred sound level for neurons with non-monotonic rate level functions.

## Computational model

**Single neuron model.** The single unit model used in this study was based on the model published by Bendor 2015 [20]. A conductance-based leaky integrate-and-fire model was simulated using MATLAB using the following equation, using parameters obtained from Wehr and Zador 2003 (Table 1) [22]:

$$V_{t+1} = -\frac{dt}{C}\left[g_e(t)(V_t - E_e) + g_i(t)(V_t - E_i) + g_{rest}(t)(V_t - E_{rest}) + V_t + \sigma_s\omega_n\sqrt{\Delta t}\right]$$

Each acoustic pulse was simulated as the summation of 10 excitatory and 10 inhibitory synaptic inputs [20], each temporally jittered (Gaussian distribution, $\sigma = 1$ ms). Each synaptic input was modelled as a time-varying conductance fit to an alpha function:

$$\alpha(t) = A(t)te^{-\frac{t}{\tau_s}}$$

A white gaussian noise term $\sigma_s\omega_n\sqrt{\Delta t}$ was added to the equation to generate a spontaneous rate of approximately 4 spikes per second in the simulated neuron.

When simulating neurons without short-term plasticity, $A$ was determined by the excitatory or inhibitory input parameter and stayed constant throughout the simulation. This amplitude ranged between 0 to 6nS for excitatory inputs and 0 to 12nS for inhibitory inputs, as in Bendor 2015 [11]. A synaptic input delay was added to simulate the delay between

**Table 1. Fixed model parameters.**

| | | |
|---|---|---|
| Membrane capacitance | $C$ | 0.25nF |
| Leak membrane conductance | $g_{rest}$ | 25nS |
| Excitatory reversal potential | $E_e$ | 0mV |
| Inhibitory reversal potential | $E_i$ | -85mV |
| Alpha function time constant | $\tau_s$ | 5ms |
| Synaptic input delay | | 10ms |
| I-E delay | | 5ms |
| Simulation timestep | $\Delta t$ | 0.1ms |
| Scale of noise | $\sigma_s$ | 10mV$s^{-1}$ |
| Gaussian noise | $\omega_n$ | [-1 :1] |

peripheral auditory system and auditory cortex, and whereas in the previous study the temporal delay between excitatory and inhibitory inputs (I-E delay) was a variable, in this study it was fixed at 5 ms. In our model, an action potential occurred whenever the membrane potential of the model neuron reached a threshold value $V_{th}$. After the action potential, the potential was reset to a value $E_{rest}$ below the threshold potential, $E_{rest} < V_{th}$.

**Short-term plasticity: Depression.** In order to introduce short-term plasticity in the model we regarded the probability of presynaptic release $P_{rel}$ as a dynamic variable depending on the input stimuli (acoustic pulse trains) [23,24,68]. In the absence of presynaptic activity, the release probability decays exponentially back to its initial value $P_0$ with the following equation:

$$\tau_P \frac{dP_{rel}}{dt} = P_0 - P_{rel}(t)$$

Immediately after each stimulus input the release probability is reduced.

$$P_{rel}(t) \rightarrow (1 - A_D) * P_{rel}(t)$$

$$A(t) = A(0) * P_{rel}(t)$$

Where $A_D$ controls the amount of depression and $A(t)$ is the amplitude of conductance input at time $t$. Modelling synaptic depression consisted thus of 4 parameters: the recovery time constants for both excitatory and inhibitory synapses ($\tau_{pE}, \tau_{pI}$) ranging from 50 to 200ms, and the depression factor $A_{DE}$ and $A_{DI}$ ranging from 0 to 0.5. $P_0$ in this model was equal to 1. These values were consistent with intra-cellular recordings in previous studies [24,25].

**Short-term plasticity: Facilitation.** Short-term facilitation was added to the model using a similar model to that of short-term depression. In the case of facilitation, $A_D$ varies between -0.5 and 0. Therefore, the probability of release $P_{rel}(t)$ Increases after each stimulus input, then decays back to the initial value. When modelling facilitation $P_0$ was equal to 0.5 so that the resulting amplitude of conductance remained comparable to that of short-term depression.

**Spike-frequency adaptation.** We modelled spike-frequency adaptation by including an addition current in the model.

$$V_{t+1} = -\frac{dt}{C}\left[g_e(t)(V_t - E_e) + g_i(t)(V_t - E_i) + g_{rest}(t)(V_t - E_{rest}) + g_{sra}(t)(V_t - E_K) + V_t + \sigma_s \omega_n \sqrt{\Delta t}\right]$$

Where $g_{sra}$ is the spike-frequency adaptation conductance modelled as a $K^+$ conductance [68]. When activated, this will hyperpolarize the neuron, slowing any spiking that may be occurring. The conductance relaxes to zero exponentially with the time constant $\tau_{sra}$ through the following equation:

$$\tau_{sra} \frac{dg_{sra}}{dt} = -g_{sra}$$

Whenever the neuron fires a spike, $g_{sra}$ is increased by an amount $\Delta g_{sra}$, causing the firing rate to adapt in a sequence of steps in relation to the neurons spiking activity.

## Data analysis

**Classification of neurons, synchrony.** Two tests were used to determine whether a neuron was Sync or nSync: Vector strength (VS) and rate response. **Vector strength** (VS) was

calculated for each repetition rate from 8 to 48Hz with the following equation:

$$VS = \frac{1}{N} \sqrt{\sin(\frac{2\pi t^{(n)}}{IPI})^2 + \cos(\frac{2\pi t^{(n)}}{IPI})^2}$$

$$RS = 2 * N * VS^2$$

Where $N$ is the number of spikes, $t^{(n)}$ is the time of $n^{th}$ pulse and $IPI$ the interpulse interval. If vector strength was significant (Rayleigh statistic $RS > 13.8$) and above 0.1 for three consecutive repetition rates, and if the rate response was also considered significant (average discharge rate 2 s.d. above the mean spontaneous rate and an average of more than 1 spike per stimulus), then the neuron was considered Sync. If the rate response was significant but the neuron did not pass the synchrony criteria, it was considered nSync. In our dataset 107/274 neurons were classified as Sync.

**Classification of neurons, monotonicity.** The monotonicity of the discharge rate for a given repetition rate was determined by calculating the Spearman correlation coefficient ($\rho$) for stimuli spanning from 8 to 48Hz. If coefficient was larger than 0.8 and statistically significant (p-value $< 0.05$) the neuron was considered positive monotonic. If the coefficient was smaller than -0.8 and statistically significant, the neuron was considered negative monotonic. Neurons satisfying neither of these criteria were considered non-monotonic. These three classification methods applied to both real and simulated neurons. In our dataset of real neurons, we found 126/274 monotonic neurons.

**Classification of neurons, Sync+ and Sync- neurons.** Based on the two classification criteria, we classified 25 Sync+ and 26 Sync- neurons with significant stimuli-driven responses, and 56 SyncNM (Non-monotonic) neurons.

**PSTH.** Individual peri-stimulus time histograms (PSTHs) were calculated by convolving a Gaussian kernel ($\sigma = 10$ms) with a neuron spike train. The population PSTH was calculated as a mean of individual PSTHs.

## Supporting information

**S1 Fig. Adaptation to stimulus pulse trains in real Sync+ and Sync- neurons.** For all neurons, we calculated the difference in normalized firing rate between the first and last acoustic pulse for a given stimulus. (a.) For Sync- neurons, this difference was significative for all repetition rates (Wilcoxon signed rank test, $P << 0.001$) with the exception of 8Hz (Wilcoxon signed rank test, $P = 0.10$). For Sync + neurons, this difference was significative for repetition rates equal or larger than 16Hz, with the exception of 40Hz (8Hz; $P = 0.71$. 12Hz; $P = 0.06$. 16Hz; $P = 0.007$. 20Hz; $P = 0.04$. 24Hz; $P = 0.009$. 28Hz; $P = 0.006$. 32Hz; $P = 0.002$. 36Hz; $P = 0.01$. 40Hz; $P = 0.07$. 44Hz; $P = 0.03$. 48Hz; $P = 0.04$). (b.) We then compared this difference between Sync+ and Sync- neuron populations (n = 25 and n = 26 respectively). This difference was significant for repetition rates above 20 Hz. (Wilcoxon rank-sum test. 8Hz; $P = 0.37$. 12Hz; $P = 0.61$. 16Hz; $P = 0.12$. For higher repetition rates $P << 0.01$).
(PDF)

**S2 Fig. Real Sync+ (A.) and Sync- (B.) neuron responses to stimulus pulse trains.** For all neurons, the average number of spikes were extracted at each acoustic pulse for all repetition rates. The responses were then normalized by average discharge rate of the neuron during stimulus presentation. Real data (grey), linear fit (red) first degree exponential fit (blue).
(PDF)

**S3 Fig. Fitted model coefficients to adaptation during stimulus presentation.** (A, B) linear model coefficients with 95% confidence intervals. Stronger negative values of p1 indicate stronger depression during stimulus presentation. (C.) R-squared fit of data to linear model. (C, D) exponential model coefficients with 95% confidence intervals. Stronger negative values of b indicate a steeper curve to the exponential model, indicating a fast adaptation followed by a flat response. Positive values of b indicate no adaptation or facilitation.
(PDF)

**S4 Fig. Onset response amplitude relative to strength of adaptation.** Average onset response at time constants $\{\tau_{pE} = 0.15, \tau_{pI} = 0.10\}$ for different values of $A_{DE}$ and $A_{DI}$. Onset response amplitude did not vary with strength of adaptation.
(PDF)

**S5 Fig. Monotonicity of real and simulated neurons.** Comparison between simulated and real neuron population PSTH for Sync+ (A; n = 30, B; n = 25) and Sync- (C; n = 30, D; n = 26) neurons.
(PDF)

**S6 Fig. Model robustness regarding Vector Strength.** Vector strength in relation to noise amplitude in Sync+ (A.) and Sync- (B.) neurons, and in relation to temporal jitter in Sync+ (C.) and Sync- (D.) neurons. Average vector strength (E, F.) across different values for recovery time constants $\{\tau_{pE}, \tau_{pI}\}$ ranging between 0.06 and 0.20s for a given value of $\{A_{DE}, A_{DI}\}$. Vector strength is maintained for E strength above 2nS and is minimally affected by IE ratio in both scenarios where model parameters produced Sync+ or Sync- neurons.
(PDF)

**S7 Fig. Comparison of ISI after stimulus onset.** ISIs between the first four spikes were compared to determine the presence of SRA for real Sync+ (A,B) and Sync- (C,D) real neuron populations for all individual trials across all neurons (n = 250 and n = 260 respectively). All four distributions had a non-zero median (KS test, P < 0.05). For Sync+ neurons, the median difference between first and second ISI was 0.59s (A.) and was 1.21ms for the median difference between first and third ISI (B.). For Sync- neurons, the median difference between first and second ISI was 0.33s (C.) and was 0.24ms for the median difference between first and third ISI (D.).
(PDF)

**S8 Fig. Monotonicity and adaptation in individual neurons.** (A). correlation between adaptation and firing rate. Distribution of strength of adaptation near onset (B.) and at the middle of stimuli duration (C.) real Sync- neurons showed significant depression between the first and second (median = 0.73, t-test, P<< 0.001) and between first and third pulse (median = 0.90, P << 0.001) (B.), but not between 2nd and 5th pulse nor between 5th and 8th pulse (median = -0.12, P = 0.33 and median = 0.07, P = 0.51 respectively.) (C.). real Sync + neurons showed no significant depression between 1st and 2nd pulses and between 1st and 3rd pulses respectively (median = 0 for both, t-test, P = 0.12 and P = 0.25 respectively) nor at the later stages of stimuli presentation between 2nd and 5th pulse (median = -0.33, p value = 0.31), and between 5th and 8th pulse, (median = 0.07 p value = 0.54).
(PDF)

**S9 Fig. Puretone responses.** (A.) Average firing rate for simulated Sync+ and Sync- responses to pure tones. (B.) Average firing rate for real Sync + (n = 25) and Sync- (n = 26) neurons to pure tones. (C, D) Effect on varying excitation strength for simulated Sync+ (C.) and Sync-

(D.) responses.
(PDF)

**S10 Fig. Puretone responses and SFA.** Puretone responses in simulated Sync+ (A.) and Sync-
(B.) neurons. SFA was introduced to our model with values ranging between 10 and 50nS (see
methods). Stronger SFA reduced both onset and sustained responses on Sync+ model neurons
but did not affect Sync- neurons. (C.) Average of median spike times during stimuli presenta-
tion for simulated neurons with different values of adaptation amplitude $A_D$.
(PDF)

## Acknowledgments

The authors thank Catherine Perrodin and James Cooke for comments and suggestions
related to this manuscript.

## Author Contributions

**Conceptualization:** Jong Hoon Lee, Daniel Bendor.

**Data curation:** Jong Hoon Lee, Xiaoqin Wang, Daniel Bendor.

**Formal analysis:** Jong Hoon Lee.

**Investigation:** Jong Hoon Lee.

**Methodology:** Jong Hoon Lee.

**Project administration:** Daniel Bendor.

**Resources:** Jong Hoon Lee, Daniel Bendor.

**Software:** Jong Hoon Lee, Daniel Bendor.

**Supervision:** Daniel Bendor.

**Validation:** Jong Hoon Lee, Daniel Bendor.

**Visualization:** Jong Hoon Lee.

**Writing – original draft:** Jong Hoon Lee.

**Writing – review & editing:** Xiaoqin Wang, Daniel Bendor.

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
