## [Decision Letter · Decision Letter 0]

10 Sep 2019

Dear Dr Lee,

Thank you very much for submitting your manuscript 'The Role of Adaptation in Generating Monotonic Rate Codes in Auditory Cortex' for review by PLOS Computational Biology. Your manuscript has been fully evaluated by the PLOS Computational Biology editorial team and in this case also by independent peer reviewers. The reviewers appreciated the attention to an important problem, but raised some substantial concerns about the manuscript as it currently stands. While your manuscript cannot be accepted in its present form, we are willing to consider a revised version in which the issues raised by the reviewers have been adequately addressed. We cannot, of course, promise publication at that time.

Please ensure that the github repository is accessible to the reviewers and contains all the files.

Sincerely,

Maria N. Geffen

Guest Editor

PLOS Computational Biology

Samuel Gershman

Deputy Editor

PLOS Computational Biology

[LINK]

Reviewer's Responses to Questions

**Comments to the Authors:**

Reviewer #1: Please see attachment.

Reviewer #2: This study uses computational modeling and analysis of previously published data to test how well a relatively simple model of synaptic depression can explain rate coding properties of neurons in auditory cortex for periodic sound stimuli. Previous models based on the relative strength and latency of excitatory and inhibitory inputs have been able to explain rate coding, but only with biologically implausible parameters. In particular, monotonic decreasing responses (“Sync-”) for higher temporal rates are not well-described. The current study uses simulations to demonstrate that a neuron in which excitatory inputs undergo synaptic depression reproduces these monotonic decreasing responses, including some detailed temporal dynamics of the neural response.

This study is elegant in exploring how a simple mechanism can explain complex neural response properties, and the simulations appear to have been implemented carefully and thoughtfully. Moreover the demonstrations of robustness are compelling. There are, however, concerns about how conclusive the results are in supporting the specific role of synaptic depression. In addition, some aspects of the authors’ thinking are not clear. For example, how does the final section on multiplexing relates to the question of mechanisms that produce rate coding?

MAJOR CONCERNS.

1. (LL. 168-192) “Different mechanisms…” The manuscript explores two possible alternative mechanisms for producing rate coding, which is helpful, but it seems somewhat limited if the authors really want to claim, as they do in the abstract, that STD “best explains” the rate coding effects. For example, what about inhibitory feedback? Or slow Ih currents? Where exactly in the circuit are the authors postulating that the STD takes place? Testing a broader set of models would be great. But at the very least, the authors should provide a more comprehensive review of possible mechanisms, particularly at the network level. The authors might consider, for example, work by the Geffen lab related to stimulus specific adaptation (Natan et al. 2015). It is also not clear how the current model difference from Gao et al. 2016.

2. (LL. 241-267). The section on multiplexing is interesting, but does it have anything to do with the initial question of mechanism producing non-phase locked responses? It seems out of place and somewhat trivial in how it is tested. Could it not work? It also seems related to previous work on color opponency, which is not mentioned. Currently an outsized portion of the Discussion focuses on this topic, which is strange, given the nominal focus of the manuscript. It would be help if there were a clearer connection to the rest of the study or perhaps if the study were introduced differently.

LESSER CONCERNS

L. 55. The authors might want to specifically mention the somatosensory system by name here, since quite a few of the relevant concepts were developed there.

L. 67. “repetition rate ranged beyond acoustic flutter” It is not clear what exactly the problem is here. Can the authors clarify what is different about this model vs. the current STD model and what it fails to account for?

L. 71. The authors may be interested in a recent preprint from the David lab (Lopez Espejo et al 2019) showing a trend toward stronger synaptic depression in excitatory vs. inhibitory inputs in auditory cortex during natural sound processing. Is this relevant to the predominant adaptation of excitatory inputs required for their model.

LL. 86-92. Something is not clear in the logic of these sentences. Does the I/E model fail to produce significant sync- effects, even with biologically implausible parameters? Or only if the parameters are forced to be realistic?

L. 118. Please cite the source for the STD model in the results. Currently it’s only mentioned in the Methods.

L. 126. “Naturalistic range” Please provide a citation.

L. 129. The monotonicity index is introduced with no definition, and it is not clear exactly how this relates to the “sync+” and “sync-“ categories even after reading through the methods. In particular, how is “monotonicity” related to synchronous vs. asynchronous responses? It would help if earlier in the results there was more explication of how rate/temporal coding is analyzed before getting in to the details of the various models.

L. 131. Please provide units for tau parameters.

LL. 157-58. This observation suggests that Sync- neurons might show differences in their phase-locking relative to Sync+ neurons. Is this the case?

L. 198. Related to spontaneous rate, is there any relationship between spont rate and type of neuron, sync+ vs. sync-? STD has been hypothesized to be more prominent in neurons with low spont rates (thus not in an adapted state prior to activation).

LL. 217-239/Fig. 10. This section is quite interesting in relating rate coding properties to fine-scale temporal dynamics. While the sync+/- responses roughly parallel each other in the real vs. simulated data, there are substantial differences. E.g., in the simulation, there is a slow rise time of the sync+ response and complete absence of a sustained response for sync-. Conversely, there is likely to be substantial variability within the actually neural data. Are there parameters in the model that can be varied to produce the variability observed in the real neurons?

L. 226. Should “differ with” be “differ from” ?

LL. 307-316. Are the any observations of synchronization outside of primary auditory cortex? Or are the authors arguing that R vs. A1 provide a hierarchy parallel to S1 and S2?

LL. 317-324. These caveats seem relevant, but very limited in their focus. Neural networks are quite complicated and it seems like network effects as well as other single cell mechanisms (like Ih currents) could play a role in producing rate responses. It seems like the topic of alternative models deserves a lot more attention in the Disucssion.

L. 368. Is code available for the simulations? Or can some more details be provided about how the simulations were executed?

L. 371. Is the choice of 10 E and I inputs important? Or does the N here simply scale inversely with the strength of the synapses to produce the desired behavior?

Fig. 10c. The colors are a big difficult to distinguish and the asterisks are quite small.

General note on figures: The current color maps do not print well in black and white.

Supplemental figures: Please indicate in legends when simulated vs. real data are used.

Reviewer #3: Overall, this is a nice paper, where the authors extend a previously described integrate and fire framework to model positive/negative monotonic rate functions. They begin by showing that the model as previously used is only capable of modeling monotonically increasing functions, as monotonically decreasing functions can only be achieved using biologically unrealistic excitatory/inhibitory parameters. The rest of the paper is based around an implementation of short-term depression in the model, and how it can produce the desired monotonic functions.

I have no issues with the modeling framework presented in the paper - the integrate and fire component of the model has been previously used and validated, and the synaptic depression extension follows standard approaches (due to Tsodyks, Abbott, etc). My main criticism is more to do with the writing, interpretation, and generalizability. Parts of the manuscript are not particularly clear, and the motivation is lacking. The modeling also seems specifically tailored to stimulus-synchronizing monotonically tuned neurons. It is not clear what fraction of the cortical population this represents, and I was left constantly asking myself about how generalizable the modeling framework is, with respect to the huge diversity of cell-types and responses present in the cortex. In addition, there are twelve figures. Many of these seem somewhat superfluous to the narrative (and are not referred to in the text) and could perhaps be moved to the supplementary information. Others could perhaps be combined.

Specific Comments

1) Line 50. With regards to the comment about “brain regions downstream from auditory cortex” - It's not clear what you are referring to here - can you give a specific example?

2) Lines 55-60. I'm a little confused about the motivation here, specifically with the final question about "how could the brain generate these types of neural representations". What “types” are you referring to - are you simply asking how can the brain generate positive/negative monotonic functions? This needs to be made more explicit.

3) Related to the comment on line 55, about “rate coding taking the form of positive and negative monotonic tuning”. What about non-monotonic tuning (also somewhat ubiquitous in cortex) - would you not call this rate coding?

4) Line 139. Figure 4 isn't really talked about in the text. If it's just a passing reference then maybe make it a supplemental figure?

5) Similar to previous comment, but with Figure 5. Panels E and F are the only panels that are explicitly referred to in the text. If these are not crucial to the story, then perhaps consider placing in supplement.

6) In the model robustness section, it’s clear what has been done but not why. Could you add a few sentences at the beginning of the section to address the biological and mathematical relevance and necessity of adding noise in this way? What is actually being achieved?

7) Could Figures 6 and 7 be combined into one figure looking at model robustness?

8) The section on “pure tone responses” comes a little out of left field. It’s not made particularly clear why this section is necessary, or why it is related to the rest of the paper.

9) The demultiplexing section is an interesting addition, but it doesn’t seem clear that this idea is really biologically relevant. In practice, a sound is made up of *lots* of acoustic features (of which, things like frequency, level, repetition rate) are only a few. Is there *any* reason to believe, or any evidence to suggest, that every potential acoustic feature has to have monotonic tuning?

10) Can the authors provide additional simulations that suggest that summing or subtracting firing rates can lead to demultiplexing of more than two acoustic features?

11) What fraction of the data set are the Sync+ and Sync- neurons (i.e how many non-monotonic neurons are there)? If this model is only suitable for a sub-set of cortical neurons (the monotonic ones), then the implications of biological differences in the circuitry underlying non-monotonic neurons needs to be discussed.

12) Similarly, the neuron’s studied seem to be only those that synchronize to the stimulus. What fraction are these?

13) Could Figures 11 and 12 be combined into one figure?

Minor Points.

1) Line 11: missing article? "represented temporally by the phase-locked..."

2) Line 29: extra s? "mechanisms that generate"

3) Line 31: pluralize code? "generate monotonic rate code"

4) Line 193: extra s in "neurons populations".

**Have all data underlying the figures and results presented in the manuscript been provided?**

Reviewer #1: No: The authors have provided a Github repository, which presumably contains the code to reproduce the data from their simulations. They could consider creating a DOI linked to the version of the code that was used in the study for posterity (https://guides.github.com/activities/citable-code/). The authors have indicated that all data are fully available, but I cannot see a link to the simulation and electrophysiological datasets.

Reviewer #2: Yes

Reviewer #3: No: Source code repository has been provided, data repository has not (although full availability has been indicated).

PLOS authors have the option to publish the peer review history of their article (what does this mean?). If published, this will include your full peer review and any attached files.

Reviewer #1: No

Reviewer #2: No

Reviewer #3: No

---

## [Decision Letter · Decision Letter 1]

11 Dec 2019

Dear Dr Lee,

Thank you very much for submitting your manuscript, 'The Role of Adaptation in Generating Monotonic Rate Codes in Auditory Cortex', to PLOS Computational Biology. As with all papers submitted to the journal, yours was fully evaluated by the PLOS Computational Biology editorial team, and in this case, by independent peer reviewers. The reviewers appreciated the attention to an important topic but identified some aspects of the manuscript that should be improved.

We would therefore like to ask you to modify the manuscript according to the review recommendations before we can consider your manuscript for acceptance. Your revisions should address the specific points made by each reviewer and we encourage you to respond to particular issues Please note while forming your response, if your article is accepted, you may have the opportunity to make the peer review history publicly available. The record will include editor decision letters (with reviews) and your responses to reviewer comments. If eligible, we will contact you to opt in or out.raised.

- Supporting Information uploaded as separate files, titled 'Dataset', 'Figure', 'Table', 'Text', 'Protocol', 'Audio', or 'Video'.

We hope to receive your revised manuscript within the next 30 days. If you anticipate any delay in its return, we ask that you let us know the expected resubmission date by email at ploscompbiol@plos.org.

Sincerely,

Maria N. Geffen

Guest Editor

PLOS Computational Biology

Samuel Gershman

Deputy Editor

PLOS Computational Biology

[LINK]

Editor's comments:

The manuscript is much improved and we ask that you consider the comments from Reviewer 1 in finalizing the text. In addition, please make the data available through an online database as per the journal policy. The manuscript states that the dataset has been published as: Differential neural coding of acoustic flutter within primate auditory cortex, Daniel Bendor & Xiaoqin Wang Nature Neuroscience volume 10, pages763–771(2007). However, we did not find a link in the publication to an online database.

Reviewer's Responses to Questions

**Comments to the Authors:**

Reviewer #1: The manuscript has been significantly improved. I have only a couple of minor points:

1. The title of Figure 7, I think, is misleading. I HIGHLIGHT the problematic parts: "Adaptation between INDIVIDUAL synaptic inputs for REAL neurons and different models". I don't think that panel B, which deals with the real neurons, has anything to do with individual synaptic inputs. It shows the strength of adaptation measured as a ratio between the 1st and 2nd or 3rd pulses. The connection to individual synaptic inputs is a conjecture here, as their adaptation was not measured but inferred. Please replace the title with one that would accurately describe what the figure actually shows.

2. The new paragraph starting on line 329 is poorly written. Please check the grammar throughout, starting from line 329.

These two minor issues notwithstanding, i think that this is an interesting and well-executed study and I recommend that it should be accepted.

Reviewer #2: The authors have done a good job addressing concerns raised during the initial review.

Reviewer #3: The authors have addressed all of my concerns. I'm happy to recommend acceptance.

**Have all data underlying the figures and results presented in the manuscript been provided?**

Reviewer #1: Yes

Reviewer #2: Yes

Reviewer #3: No: The authors have not provided an explanation as to their "exceptional situation" (per journal policy), regarding making their data only available on request.

PLOS authors have the option to publish the peer review history of their article (what does this mean?). If published, this will include your full peer review and any attached files.

Reviewer #1: No

Reviewer #2: No

Reviewer #3: No

---

## [Editor Report · Decision Letter 2]

2 Jan 2020

Dear Dr Lee,

We are pleased to inform you that your manuscript 'The Role of Adaptation in Generating Monotonic Rate Codes in Auditory Cortex' has been provisionally accepted for publication in PLOS Computational Biology.

In the meantime, please log into Editorial Manager at https://www.editorialmanager.com/pcompbiol/, click the "Update My Information" link at the top of the page, and update your user information to ensure an efficient production and billing process.

One of the goals of PLOS is to make science accessible to educators and the public. PLOS staff issue occasional press releases and make early versions of PLOS Computational Biology articles available to science writers and journalists. PLOS staff also collaborate with Communication and Public Information Offices and would be happy to work with the relevant people at your institution or funding agency. If your institution or funding agency is interested in promoting your findings, please ask them to coordinate their releases with PLOS (contact ploscompbiol@plos.org).

Thank you again for supporting Open Access publishing. We look forward to publishing your paper in PLOS Computational Biology.

Sincerely,

Maria N. Geffen

Guest Editor

PLOS Computational Biology

Samuel Gershman

Deputy Editor

PLOS Computational Biology

The revisions are fine. We thank the authors for providing the data set.

---

## [Editor Report · Acceptance letter]

7 Feb 2020

PCOMPBIOL-D-19-01000R2 

The role of adaptation in generating monotonic rate codes in auditory cortex

Dear Dr Lee,

I am pleased to inform you that your manuscript has been formally accepted for publication in PLOS Computational Biology. Your manuscript is now with our production department and you will be notified of the publication date in due course.

With kind regards,

Laura Mallard
